# Anti-Inflammatory Effects of Minor Cannabinoids CBC, THCV, and CBN in Human Macrophages

**DOI:** 10.3390/molecules28186487

**Published:** 2023-09-07

**Authors:** Esmaeel Ghasemi Gojani, Bo Wang, Dong-Ping Li, Olga Kovalchuk, Igor Kovalchuk

**Affiliations:** Department of Biological Sciences, University of Lethbridge, Lethbridge, AB T1K 3M4, Canada; esmaeel.ghasemigojan@uleth.ca (E.G.G.); bo.wang5@uleth.ca (B.W.); dongping.li@uleth.ca (D.-P.L.)

**Keywords:** inflammation, THP-1 macrophage, THCV, CBC, CBN

## Abstract

Inflammation is a natural response of the body to signals of tissue damage or infection caused by pathogens. However, when it becomes imbalanced, it can lead to various disorders such as cancer, obesity, cardiovascular problems, neurological conditions, and diabetes. The endocannabinoid system, which is present throughout the body, plays a regulatory role in different organs and influences functions such as food intake, pain perception, stress response, glucose tolerance, inflammation, cell growth and specialization, and metabolism. Phytocannabinoids derived from *Cannabis sativa* can interact with this system and affect its functioning. In this study, we investigate the mechanisms underlying the anti-inflammatory effects of three minor phytocannabinoids including tetrahydrocannabivarin (THCV), cannabichromene (CBC), and cannabinol (CBN) using an in vitro system. We pre-treated THP-1 macrophages with different doses of phytocannabinoids or vehicle for one hour, followed by treating the cells with 500 ng/mL of LPS or leaving them untreated for three hours. To induce the second phase of NLRP3 inflammasome activation, LPS-treated cells were further treated with 5 mM ATP for 30 min. Our findings suggest that the mitigation of the PANX1/P2X7 axis plays a significant role in the anti-inflammatory effects of THCV and CBC on NLRP3 inflammasome activation. Additionally, we observed that CBC and THCV could also downregulate the IL-6/TYK-2/STAT-3 pathway. Furthermore, we discovered that CBN may exert its inhibitory impact on the assembly of the NLRP3 inflammasome by reducing PANX1 cleavage. Interestingly, we also found that the elevated ADAR1 transcript responded negatively to THCV and CBC in LPS-macrophages, indicating a potential involvement of ADAR1 in the anti-inflammatory effects of these two phytocannabinoids. THCV and CBN inhibit P-NF-κB, downregulating proinflammatory gene transcription. In summary, THCV, CBC, and CBN exert anti-inflammatory effects by influencing different stages of gene expression: transcription, post-transcriptional regulation, translation, and post-translational regulation.

## 1. Introduction

Inflammation is a natural response of the body to tissue damage or infection caused by pathogens. It helps restore balance by protecting against pathogens and aiding in tissue repair. However, if this response becomes dysregulated, it can lead to chronic inflammation, contributing to various inflammatory conditions like cancer, obesity, sepsis, cardiovascular disorders, neurological ailments, and autoimmune diseases [1,2,3,4].

Chronic inflammation progresses through several stages. Initially, danger signals such as DAMPs (damage-associated molecular patterns) or PAMPs (pathogen-associated molecular patterns) are released from injured cells or pathogens. These signals activate immune cells that are already present in the tissue, resulting in the production of cytokines, chemokines, and proteases. Immune cells then migrate from the site of inflammation, further amplifying pro-inflammatory responses. This highly orchestrated immune response ultimately restores homeostasis. However, if the immune response is either insufficient or excessive, it can hinder the restoration of balance and prolong inflammatory processes [5]. This dysregulated immune response initiates a detrimental series of events, characterized by the infliction of local or systemic tissue damage and an excessive release of danger signals. Furthermore, the shift from acute to chronic inflammation can compromise immune tolerance, facilitating the advancement of inflammatory diseases and, in certain instances, culminating in fatality [6].

Different innate immune pattern recognition receptors such as Toll-like receptors (TLRs) expressed on immune cells, recognize DAMPs and PAMPs [7]. Lipopolysaccharide (LPS) is a constituent of the outer membrane found in Gram-negative bacteria, which can act as PAMP to initiate a proinflammatory response. Immune cells, macrophages included, identify LPS through the pattern recognition receptor Toll-like receptor 4 (TLR4) [8]. Upon recognition of LPS, TLR4 undergoes oligomerization and recruits downstream adaptors through interactions with Toll-interleukin-1 receptor (TIR) domains [9]. Different TLRs utilize various combinations of these adaptor proteins to determine downstream signaling. Notably, TLR4 is the only known TLR that employs all of these adaptor proteins [10], triggering a cascade of activation pathways including nuclear factor-κB (NF-κB), interferon regulatory factor, and MAPK pathways. This activation leads to the production of pro-inflammatory cytokines (e.g., tumor necrosis factor (TNF), interleukin-1 (IL-1), and IL-6, and chemokines) through transcriptional and post-transcriptional mechanisms [11].

Likewise, STAT-3 is a transcription factor that, upon activation, stimulates the expression of various pro-inflammatory cytokines including IL-17, IL-23, and IL-8 [12]. Of particular importance, IL-6 serves as a critical activator of STAT-3 during inflammation, and the IL-6-STAT-3 axis has emerged as a key target for designing new treatments against many inflammatory disorders.

Additionally, excessive NF-κB activation leads to increased IL-6 levels, which, in turn, can trigger Janus kinase (JAK) phosphorylation. This phosphorylation event subsequently activates STAT-3, facilitating its translocation into the nucleus and ultimately leading to an augmented release of cytokines. The activated STAT-3 also exhibits a synergistic interaction with NF-κB, causing the hyperactivation of NF-κB and further upregulation of pro-inflammatory proteins such as IL-6, Pro-TNFα, Pro-IL-1β, and COX-2 [13,14]. Therefore, downregulation of NF-kB/IL-6/JAK/STAT-3 may provide an appropriate target to control inflammatory responses.

In addition to TLRs, immune cells possess NOD-like receptors (NLRs) that specifically detect pathogenic patterns in the cytoplasm, resulting in inflammasome-mediated activation and release of pro-inflammatory cytokines. Various kinds of inflammasomes exist including NLRP1, NLRP2, and NLRP3. Among these, NLRP3 has garnered significant attention, primarily due to the close association between its activation and inflammatory disorders, obesity, diabetes, and neurodegenerative ailments [15,16]. 

The NLRP3 inflammasome, a complex protein, consists of three primary subunits: (1) the NLRP3 protein, which comprises an N-terminal pyrin domain, a centrally located NACHT domain, and a C-terminal leucine-rich repeat region (LRR) domain; (2) the apoptosis-associated spike-like protein (ASC) domain that contains two death-fold domains, an N-terminal pyrin domain and a C-terminal caspase recruitment domain (CARD); (3) the Pro-caspase-1 domain that encompasses a CARD domain [17]. 

The activation process of the NLRP3 inflammasome can be divided into two steps. The initial step, known as the priming step, involves immune cells detecting internal or external insults, leading to the activation of NF-κB and the subsequent upregulation of proinflammatory cytokines such as Pro-IL-1β and Pro-IL-18 [17]. The second step, which includes the activation of the NLRP3 inflammasome, is governed by various mechanisms proposed in the literature including K^+^ efflux, production of reactive oxygen species (ROS), and a decrease in cAMP levels [15,18,19]. Upon stimulation of the NLRP3 inflammasome, Pro-caspase-1 is activated, subsequently facilitating the conversion of Pro-IL-1β and Pro-IL-18 into their mature forms, IL-1β and IL-18.

The endocannabinoid system (ECS) encompasses a complex combination of substances that include: (1) lipidic endocannabinoids such as anandamide (AEA) and 2-arachidonoylglycerol (2AG); (2) endocannabinoid-synthesizing enzymes such as NAPE-PLC involved in AEA synthesis and DAGL involved in 2-AG synthesis; (3) endocannabinoid-hydrolyzing enzymes like FAAH and MAGL; (4) endocannabinoid-related receptors including CB1R, CB2R, TRPV1, PPARγ, and GPR55 [20].

This system is distributed throughout the body and plays various modulatory roles in different organs, influencing functions such as food intake, pain, stress, glucose tolerance, cell proliferation and differentiation, and metabolism. While CB1R and CB2R receptors are the most prominent endocannabinoid receptors, other receptors like TRPV1, GPR119, GPR55, GPR18, GPR92, and peroxisome proliferator-activated receptors (PPARs) are also involved in modulating the endocannabinoid system [21].

*Cannabis sativa*, a plant with a long history of medicinal use, contains a diverse array of pharmacologically active compounds known as phytocannabinoids, which can collectively interact with the ECS, regulating various physiological processes [22]. Phytocannabinoids represent a consistent group of monoterpenoids found in *Cannabis sativa* [23].

Among the phytocannabinoids, delta-9-tetrahydrocannabinol (THC) and cannabidiol (CBD), also known as major phytocannabinoids, have been extensively studied. Other phytocannabinoids such as tetrahydrocannabivarin (THCV), cannabinol (CBN), cannabigerol (CBG), and cannabichromene (CBC), known as minor phytocannabinoids, are also present in cannabis. Limited research exists on the anti-inflammatory properties of minor phytocannabinoids compared to the more extensively studied major cannabinoids. Therefore, the aim of this study was to investigate and elucidate the anti-inflammatory effects of three significant minor phytocannabinoids: CBC, THCV, and CBN.

## 2. Results

### 2.1. THCV, CBC, and CBN Downregulated Proinflammatory Protein Molecules 

All three cannabinoids investigated in this study exhibited mitigating effects on dysregulated Pro-IL-β and IL-1β production in LPS + ATP (LA)-stimulated THP-1 macrophages. Both 5 μM and 15 μM THCV downregulated the immature and mature forms of IL-1β. The IL-1β/Pro-IL-1β ratio served as a reliable indicator for the second phase of NLRP3 inflammasome activation. The high dose of THCV effectively mitigated the second phase of NLRP3 activation, in contrast to the lower dose of THCV (Figure 1a). Similarly, CBC significantly suppressed the production of both Pro-IL-1β and IL-1β (Figure 1b). However, unlike THCV, the lower dose of CBC diminished the activation of the second phase of the NLRP3 inflammasome. It is noteworthy that lower doses (2.5 μM and 5 μM) of CBC were utilized for Western blot, q-PCR, and ELISA experiments in comparison to THCV and CBN, which was due to the high cytotoxicity of 15 μM CBC.

Regarding CBN, 5 μM and 15 μM exhibited different effects on dysregulated Pro-IL-1β and IL-1β in LA-induced THP-1 macrophages. Only the high dose of CBN influenced the production of Pro-IL-1β and IL-1β (Figure 1c). In terms of the second phase, neither 5 μM nor 15 μM CBN could mitigate the assembly of the NLRP3 inflammasome. In other words, the changes observed in Pro-IL-1β production were reflected in IL-1β production (Figure 1).

IL-6, TNFα, and COX-2 are crucial proinflammatory proteins induced by LPS [10]. In this study, all three phytocannabinoids demonstrated the ability to mitigate the levels of IL-6 and COX-2 at the protein level. The response of IL-6 to the phytocannabinoids was found to be dose-dependent. Similarly, both doses of THCV, CBC, and CBN were effective in reducing the production of COX-2 (Figure 1). TNFα is an important proinflammatory cytokine associated with various disorders. It is initially synthesized as an inactive precursor protein called Pro-TNFα or the TNFα precursor (25 KDa), which is membrane-bound. The mature and active form of TNFα is generated through proteolytic cleavage of the Pro-TNFα by the TNFα converting enzyme (TACE), also known as ADAM17 [24]. The results showed that both doses of THCV had a mitigating effect on both forms of TNFα (Figure 2a). In contrast, CBC increased the production of Pro-TNFα in a dose-dependent manner (Figure 2b). Notably, no results were obtained for the mature form of TNFα. Similar to THCV, both doses of CBN were able to downregulate both forms of TNFα in LPS-induced THP-1 macrophages (Figure 2c). Taken together, these results indicate that THCV, CBC, and CBN inhibit the production of proinflammatory mediators.

### 2.2. Impact of THCV, CBC, and CBN on the Response of Inflammation-Associated Transcription Factors

NF-κB (Nuclear Factor-kappa B p65) is a transcription factor that plays a critical role in regulating the expression of numerous genes involved in inflammatory responses including TNFα, IL-6, and Pro-IL-1β as well as cell viability and other processes. Normally, NF-κB is localized in the cytoplasm. However, upon induction by various stimuli such as LPS, pro-inflammatory cytokines, pathogens, or intracellular stresses, the IκB subunit of NF-κB is phosphorylated, leading to its degradation. This results in the translocation of NF-κB into the nucleus, where it drives the transcription of inflammation-related proteins [25].

Both doses of CBN and THCV exhibited the ability to downregulate the phosphorylated form of NF-κB (P-NF-κB) as well as the total NFκB (t-NF-κB) (Figure 3a,c). The ratio between P-NF-κB and t-NFκB serves as a suitable indicator for NF-κB phosphorylation. The results indicate that only the lower dose of THCV mitigates the phosphorylation of NF-κB, while both doses of CBN have negative effects on NF-κB phosphorylation. However, none of the doses of CBC exhibited inhibitory effects on either the transcription or phosphorylation of NF-κB (Figure 3b).

The IL-6/JAK/STAT signaling pathway plays a crucial role in the secretion of pro-inflammatory cytokines and the regulation of various biological processes including cell proliferation, differentiation, and apoptosis. The JAK (Janus kinase) family consists of four non-receptor protein tyrosine kinase members: JAK1, JAK2, JAK3, and TYK (tyrosine kinase).

Upon the binding of IL-6 to its receptor, conformational changes occur in the receptor, leading to the receptor dimerization and bringing their associated JAKs in close proximity to each other. This event triggers the phosphorylation and subsequent activation of JAK family enzymes including TYK-2 [13,14].

Following activation, JAK enzymes facilitate the phosphorylation of specific tyrosine residues on the dimerized receptors. This phosphorylated tyrosine serves as docking sites for cytoplasmic STATs (signal transducers and activators of transcription), which are then phosphorylated and activated by JAKs. Once phosphorylated, STATs dissociate from the receptors, form dimers, and translocate into the nucleus. Inside the nucleus, they act as transcription factors, driving the expression of proinflammatory cytokines and other target genes [26].

In this study, the focus was on investigating the responses of TYK-2 and STAT-3 phosphorylation to the phytocannabinoids. Interestingly, all three phytocannabinoids examined exhibited the downregulation of STAT-3 phosphorylation and subsequent activation (Figure 4).

Both 5 μM and 15 μM doses of THCV demonstrated the downregulation of the phospho-STAT-3 (P-STAT-3) levels as well as a decrease in the ratio of P-STAT-3/STAT-3 (Figure 4a). However, the total STAT-3 (t-STAT-3) levels did not show any significant response to THCV treatment. These findings suggest that THCV influences STAT-3 activation through the mitigation of phosphorylation rather than the downregulation of t-STAT-3.

The effects of THCV on TYK-2 activation were consistent with the response observed for STAT-3 (Figure 5a), suggesting that THCV mitigates STAT-3 activation through the reduction in TYK-2 phosphorylation. Although 5 μM THCV did not affect TYK-2 phosphorylation, 15 μM THCV significantly decreased phosphor-TYK-2 (P-TYK-2) levels in the LPS-induced macrophages (Figure 5a).

The response of total-TYK-2 (t-TYK-2) to both 5 μM and 15 μM doses of THCV indicated a significant increase in the t-TYK-2 levels in the LPS-induced macrophages (Figure 5a). The P-TYK-2/TYK-2 ratio, which serves as an index for TYK-2 phosphorylation or activation, showed that both doses of THCV exerted their effects on TYK-2 activation by mitigating its phosphorylation.

Both doses of CBN downregulated the P-STAT-3 levels, t-STAT-3 levels, and the P-STAT-3/STAT-3 ratio (Figure 4c), indicating that CBN decreases STAT-3 activation through both the downregulation of t-STAT-3 translation and the decrease in phosphorylation.

In contrast to THCV, CBN significantly upregulated the P-TYK-2 levels and the P-TYK-2/TYK-2 ratio (Figure 5c). However, this phytocannabinoid showed a dose-dependent downregulation of t-TYK-2 in the LPS-induced macrophages.

The higher dose of CBC had a negative effect on P-STAT-3 and P-STAT-3/t-STAT-3, while the lower dose did not show significant effects on these proteins (Figure 4b). Conversely, the lower dose of CBC significantly downregulated the P-TYK-2 and TYK-2 levels as well as the P-TYK-2/TYK-2 ratio, whereas the higher dose did not have a significant impact on them (Figure 5b).

The activation of STAT-1 is linked to the increase in the levels of the *ARID5A* transcript, an enzyme that plays a role in stabilizing IL-6 mRNA [27]. 

Based on our Western blot results, both doses of THCV and 15 μM CBN were found to reduce the elevated level of phosphorylated STAT-1 (P-STAT-1) in the macrophages stimulated with LPS. This reduction is likely achieved by mitigating the phosphorylation of STAT-1 rather than affecting the total levels of STAT-1 (Figure 6). Taken together, these results indicate that THCV, CBC, and CBN attenuate the LPS-triggered phosphorylation/activation of transcription factors NF-κB, STAT-3, and/or STAT-1, which may play a contributing role in the tested cannabinoid-induced reduction in proinflammatory mediators.

### 2.3. THCV, CBN, and CBC Inhibit NLRP3 Inflammasome Activation

The activation of the NLRP3 inflammasome occurs in two steps. The priming step is initiated when macrophages sense endogenous or exogenous stimuli, leading to the phosphorylation of NF-κB and the subsequent transcription of proinflammatory genes such as *IL-1β*, *TNFα*, *IL-18*, *IL-6*, and *NLRP3*. The second step involves the assembly of NLRP3 inflammasome components, the conversion of Pro-caspase-1 into C-caspase-1, and the subsequent production of IL-1β.

Significantly, both 5 μM and 15 μM THCV, CBN, and both doses of CBC decreased the NLRP3 and Pro-IL-1β protein levels in LA-induced THP-1 macrophages (Figure 1 and Figure 7), suggesting the mitigation of the priming step by these phytocannabinoids.

Both doses of THCV downregulated Pro-caspase-1 (Figure 7a) whereas none of the doses of CBC downregulated the level of Pro-caspase-1 at the protein level (Figure 7b). These results suggest that the tested cannabinoids may inhibit NLRP3 inflammasome activation at both the priming and second steps.

### 2.4. All Three Phytocannabinoids Downregulate Pannexin-1 (PANX-1) Expression and Inhibit Its Activation 

PANX-1 is a transmembrane transporter that plays a role in the upregulation and activation of proinflammatory cytokines, particularly IL-1β. Upon exposure to various stimuli like LPS, macrophages increase the expression and activation of PANX-1, leading to the efflux of intracellular molecules including ATP. Extracellular ATP can then bind to purinergic receptors, particularly P2X7 receptors, on macrophages. This activation of P2X7 receptors initiates a signaling cascade, ultimately leading to the activation of the NLRP3 inflammasome, Pro-caspase-1 cleavage, and the conversion of Pro-IL-1β into IL-1β [28].

Under normal conditions, the PANX-1 channel is blocked by its cytoplasmic C-terminal region. However, suitable stimuli can induce apoptotic-related caspases such as caspase-3 and -7 to cleave the C-terminus of PANX-1, resulting in the opening of PANX-1 channels and subsequent ATP efflux [28,29].

Based on our findings, all three cannabinoids studied in this research downregulated the cleavage of PANX-1, thereby reducing PANX-1 opening and ATP efflux. As a result, the influx of ATP through the P2X7 receptor and the subsequent activation of the NLRP3 inflammasome, as reflected in IL-1β production, were decreased.

Both doses of THCV downregulated the levels of PANX-1 (45 kDa), while only the 5 μM dose of THCV inhibited the cleavage of PANX-1 (Figure 8a).

Both 5 μM and 15 μM CBN decreased the levels of PANX-1 (45 kDa) as well as the cleavage of PANX-1 (Figure 8c).

Both doses of CBC exerted their effects on PANX-1 activity by downregulating the levels of PANX-1 (45 kDa) and slightly inhibiting PANX-1 cleavage (Figure 8b). The results showed that CBC reduced the levels of PANX-1 (45 kDa) and C-PNX-1 in the LPS-induced macrophages in a dose-dependent manner (Figure 8b).

### 2.5. THCV, CBC, and CBN Modulate Transcription of Proinflammatory Genes 

To assess the correlation between the activation of NF-κB, STAT-1, TYK-2, and STAT-3 and the expression of *IL-1β*, *IL-6*, *TNFα*, *COX*-2, and other proinflammatory genes, we conducted qRT-PCR using specific primers.

The lower dose of THCV resulted in a decrease in the *IL-1β*, *P2X7*, *PANX1*, *TNFα*, *cPLA2*, and *ADAR-1* mRNA levels (Figure 9a). In contrast, the higher dose of THCV upregulated the transcription of these genes. Interestingly, THCV exhibited a dose-dependent increase in the *IL-6* and *COX-2* transcripts. The transcription of *NLRP3* was reduced in response to both doses of THCV, although the lower dose appeared to be more effective in decreasing the *NLRP3* mRNA levels.

Both doses of CBC demonstrated mitigative effects on the transcription of all genes examined in this study including cytokines and *COX-2* (Figure 9b). The lower and higher doses of CBC exhibited similar downregulatory effects on the transcription of all of the tested genes.

While the lower dose of CBN did not significantly downregulate *IL-1β*, *NLRP3*, and *IL-6* transcription, the higher dose resulted in a decrease in the transcription of these three genes. Both doses of CBN upregulated the transcription of *PANX1*, *cPLA2*, *P2X7*, and *COX-2* genes (Figure 9c).

These results suggest that THCV, CBC, and CBN differentially modulate the transcription of the proinflammatory genes examined.

### 2.6. THCV and CBC Mitigate the Secretion of IL-1β

To establish a relationship between the findings from Western blot analysis regarding the mature form of IL-1β and the secretion of IL-1β in the medium, an ELISA assay was performed.

Consistent with the impact of THCV and CBC on the mature form of IL-1β, both doses of THCV and CBC resulted in a decrease in the secretion of active IL-1β in the medium (Figure 10a,b). It is important to note that the higher doses of both phytocannabinoids were more effective than the lower dose in reducing IL-1β release.

Interestingly, CBN exhibited an increase in the level of IL-1β secretion, potentially in a dose-dependent manner (Figure 10c). It is noteworthy that this response differed from the effect of CBN on the mature form of IL-1β (Figure 1c).

### 2.7. Tested Phytocannabinoids Did Not Decrease the Viability of Macrophages 

An MTT assay was conducted to investigate the impact of different treatments on the viability of THP-1 macrophages. While the lower doses of CBC and THCV had no significant effect on the viability of LA-induced macrophages, the higher doses resulted in a notable increase in cell viability (Figure 11a,b). In contrast, neither the 15 μM nor 5 μM CBN had any significant effect on the viability of the LA-induced macrophages (Figure 11c). These results indicate that the tested cannabinoids have no cytotoxic effect on macrophages.

## 3. Discussion

This study aimed to elucidate the underlying mechanism of the anti-inflammatory properties exhibited by three lesser-known phytocannabinoids, namely THCV, CBC, and CBN.

### 3.1. The Effect of THCV on LA/LPS Induced Macrophages 

While there have been previous indications of THCV’s anti-inflammatory effects [30,31], none of these reports have provided insights into the targeted pathways. One such report showed that the combination of THCV, CBD, and CBG could potentially alleviate proinflammatory cytokines in lung epithelial cells. Additionally, their findings revealed that the effects of this combination were dependent on the administered dose and did not follow a bell-shaped curve response pattern. The bell-shaped curve response mechanism of action refers to a phenomenon observed with certain phytocannabinoids, particularly CBD. It describes the biphasic response where, at lower doses, the phytocannabinoid can produce desired effects such as anti-inflammatory properties. However, as the dosage increases beyond a certain point, the effectiveness of the phytocannabinoid may plateau or even diminish, resulting in reduced or no therapeutic benefits.

It is noteworthy that with a bell-shaped curve response, determining the optimal therapeutic dose and avoiding potential side effects may become challenging. Therefore, it may be difficult to effectively utilize active ingredients in clinical therapy due to the absence of a predictable dose–response relationship [32]. 

Romano et al. (2016) found that THCV shows a downregulatory effect on the excessive expression of nitric oxide synthase (iNOS), COX-2, and IL-1β proteins induced by LPS. Additionally, THCV counteracted the LPS-induced upregulation of CB1 receptors, while it did not affect changes in *CB2R*, *TRPV2*, or *TRPV4* mRNA expression [33]. However, they did not decipher the exact downregulatory mechanism of THCV. 

In one study, the downregulatory effects of 5 μM THCV on the transcription of TNFα, IL-6, and IL-1β in human adipose-derived stem cells was found [31]. However, like in the other studies, the mechanism behind these anti-inflammatory effects remained unknown. 

According to our findings, THCV exhibited the downregulation of IL-1β, IL-6, TNFα, and COX-2 at the protein level (Figure 1 and Figure 2), which could be attributed to the mitigative effects of THCV on various stages of gene expression such as transcription, post-transcriptional gene regulation, translation, and/or post-translational modifications. 

As mentioned before, the activation of NF-κB triggers the transcription of numerous proinflammatory proteins including Pro-IL-1β, IL-6, Pro-TNFα, and COX-2. 

Our study demonstrated that both 5 μM and 15 μM THCV effectively inhibited the activation of NF-κB (Figure 3). To explore whether the downregulatory effects of THCV on the cytokines and COX-2 are connected to the inhibition of NF-κB and subsequently to the transcription of these proinflammatory genes, we performed qRT-PCR.

Interestingly, while both 5 μM and 15 μM THCV led to the downregulation of p-NF-κB, only the 5 μM THCV exhibited the downregulation of Pro-IL-1β and Pro-TNFα transcription (Figure 9). This suggests that the regulatory effect of 5 μM THCV on the Pro-IL-1β and Pro-TNFα levels may be mediated, at least in part, through the downregulation of their transcription, which are probably mediated through the inhibition of NF-κB. 

In contrast to the modulatory effects of THCV on the protein levels of IL-6 and COX-2, both 5 μM and 15 μM THCV increased the transcription of these two proinflammatory proteins (Figure 9). This implies that THCV may act through post-transcriptional, translational, and/or post translational mechanisms to regulate the IL-6 and COX-2 levels in macrophages stimulated by the LPS.

To examine the possible effects of THCV on the second phase of NLRP3 inflammasome activation, the response of some genes involved in this process at transcription, translation, and post-translational modification levels was studied. 

In macrophages, the mere presence of the second phase of NLRP3 inflammasome activators is not sufficient to trigger inflammasome activation. Instead, a priming signal is necessary. This priming signal can come from ligands for toll-like receptors (TLRs), NLRs such as NOD1 and NOD2, or cytokine receptors. These priming stimuli activate the transcription factor NF-κB, which in turn upregulate NLRP3 and Pro-IL-1β. Under normal conditions, the concentration of NLRP3 is not sufficient to initiate inflammasome activation and Pro-IL-1β is not expressed at all. It is noteworthy that priming signals do not seem to affect the protein levels of ASC and Pro-caspase-1 [34].

NLRP3, a key component of the NLRP3 inflammasome, was downregulated at both the transcriptional and translational levels by both doses of THCV (Figure 7a). Since NF-κB is involved in the transcription of NLRP3, the modulatory effects of THCV on NLRP3 transcription, at least in part, are mediated through the mitigation of this transcription factor. 

In addition to NF-kB, STAT-3 could bind to the NLRP3 promoter to promote H3K9 acetylation and thereby NLRP3 transcription. This suggests that STAT-3 activation plays a role in driving the expression of the NLRP3 gene [35]. 

Since both doses of THCV diminished the activation of STAT-3, the reduction in STAT-3 activation by THCV may also take part in the downregulation of NLRP3. 

PANX-1 is a transmembrane protein that forms channels and is expressed in a variety of cell types, macrophages included. In its resting state, the PANX-1 channel is inhibited by its own C-terminal tail located in the cytoplasmic side [29]. 

During apoptosis, caspase-3 and -7 cleave PANX-1 at its C-terminus, leading to the opening of the PANX-1 channel and increased membrane permeability [29,36]. 

It has been demonstrated that PANX-1 channels are associated with the assembly and activation of the NLRP3 inflammasome during apoptosis [37]. In line with this, the treatment of macrophages with the PANX-1 inhibitor carbenoxolone or the PANX-1 inhibitory peptide 10PANX has been shown to suppress canonical NLRP3 inflammasome activation [28]. 

Our results indicate that, at least in part, the modulative effects of THCV on the second step of NLRP3 inflammasome activation is mediated through the downregulation of PANX1 transcription, translation, and cleavage (Figure 8 and Figure 9), which in turn, result in a negative impact on the production of IL-1β. It should be mentioned that only 5 μM THCV downregulated the cleavage and thereby opening of the PANX-1 protein, which is probably mediated through the downregulation of caspase-3 and -7.

Accordingly, one study found that pre-treatment of human adipose-derived mesenchymal stem cells (ASC) with THCV could downregulate caspase-3 production [31].

The activation of the NLRP3 inflammasome is associated with certain purinergic receptors including P2X7. The P2X7 receptor is a ligand-gated ion channel that modulates the efflux of intracellular potassium (K^+^) [38]. Extracellular ATP, which may arise from excessive PANX-1 activation, stimulates the P2X7 receptor, leading to elevated intracellular potassium levels and subsequent NLRP3 inflammasome activation. In the context of LPS-induced macrophages, our results demonstrate that 5 μM THCV downregulated P2X7 transcription (Figure 9). These findings suggest that a portion of the mitigating effects of 5 μM THCV on the assembly of NLRP3 is mediated through the downregulation of P2X7. 

In line with our results, it has been demonstrated that potassium efflux is a key factor in the inhibitory effects of cannabidiol (CBD) on NLRP3 inflammasome activation in THP-1 monocytes, which may be mediated through ATP-induced P2X7 activation [38]. 

The results obtained from the Western blot analysis of NLRP3, Pro-caspase-1, and Pro-IL-β (Figure 1 and Figure 7) indicate that both doses of THCV exhibited the ability to downregulate the initial phase of NLRP3 inflammasome activation in macrophages stimulated with LA, which is probably mediated through the inhibition of NF-κB activation. 

Furthermore, we observed that THCV exhibited inhibitory effects on the activation of STAT-3, which may be attributed to its potential modulation of the IL-6/TYK-2 signaling axis (Figure 4 and Figure 5), resulting in the mitigation of cytokine storms. This finding aligns with previous research demonstrating that major cannabinoids such as THC and CBD can mitigate cytokine storms by downregulating the IL-6/TYK-2/STAT-3 pathway [13].

In summary, 15 μM THCV exhibited greater efficacy compared to 5 μM in mitigating all three cytokines and the COX-2 levels. However, its mechanism of action appears to involve post-transcriptional, translational, and/or post-translational rather than transcriptional gene regulation. In agreement with this, we found that 15 μM THCV downregulated the mature form of IL-1β by inhibiting the second step of NLRP3 inflammasome activation. This is further supported by the levels of IL-1β/Pro-IL-1β, which serve as reliable indicators of the second step. In contrast, 5 μM mainly exerted its effects through downregulating the transcription. 

STAT-1 is a transcription factor involved in proinflammatory responses. Its activation is dependent on phosphorylation, which is crucial for its ability to regulate gene expression. This phosphorylation process is associated with the activation of JAK1 and TYK-2, which occurs following the stimulation of the TLR4 receptor by LPS [39,40,41]. 

Phosphorylation of STAT-1 at Thr749 enhances the stability of IL6 mRNA by activating the transcription of the gene encoding ARID5A [27]. Our results indicate that THCV can mitigate the TYK-2/STAT-1 axis, suggesting an additional mechanism through which THCV may inhibit the production of IL-6 (Figure 9).

### 3.2. THE Effect of CBC on LA/LPS Induced Macrophages 

CBC is one of the most abundant phytocannabinoids alongside THC, CBD, and CBN [42,43]. Like THC and CBD, CBC is directly synthesized from cannabigerolic acid and shares a common 3-pentylphenol ring [44]. The therapeutic potential of CBC has been demonstrated in several preclinical studies [45,46].

In addition to CB1R and CB2R, CBC also interacts with TRPA1 and adenosine receptors. CBC influences cellular endocannabinoid reuptake by suppressing the function of MAGL. CBC has demonstrated effectiveness in downregulating inflammatory responses in macrophages. It has been observed that CBC’s anti-inflammatory effects are mediated by its agonistic activity on TRPA1, resulting in reduced levels of intracellular nitric oxide (NO) and IFNγ. CB1R-mediated signaling appears to be involved in CBC’s activity [47]. However, Udoh et al. (2019) demonstrated the agonistic effects of CBC on CB2R, which, given the established role of CB2R in mitigating inflammatory responses, may provide an alternative mechanism to explain CBC’s anti-inflammatory effects [48,49]. 

Our findings revealed that both concentrations of CBC mitigated the level of IL-6, Pro-IL-1β, and COX-2 proteins while not exerting any modulation effect on the level of Pro-TNFα (Figure 1 and Figure 2). Unlike our findings, Romano et al. (2013) reported that CBC did not reduce the increased IL-1β and COX-2 levels in LPS-induced macrophages. It is important to note that their experimental parameters including the CBC dosage, LPS duration, and concentration differed from those in our present study [50]. To investigate whether these regulatory effects of CBC occurred at the transcriptional level, qRT-PCR was performed. The qRT-PCR analysis demonstrated that both doses of CBC resulted in a reduction in the transcription of all of the examined genes in the LPS-induced THP-1 macrophages (Figure 9). The consistency between the qRT-PCR results and the Western blot results for the IL-1β, IL-6, and COX-2 mRNAs/proteins suggests that CBC may exert its inhibitory effects on these proteins primarily through suppressing their transcription. Interestingly, despite CBC’s inhibitory effects on TNFα transcription (Figure 9), it exhibited a stimulatory impact on Pro-TNFα protein levels (Figure 2), indicating that CBC may influence the post-transcriptional processes of Pro-TNFα to alter its response.

While NF-κB is a crucial transcription factor involved in regulating the transcription of numerous proinflammatory proteins, it appears to have a minor role in the inhibitory effects of CBC on the transcription of *IL-β*, *TNFα*, *IL-6*, *COX-2*, and other genes. This is evidenced by the fact that none of the CBC doses exhibited inhibitory effects on NF-κB activation (Figure 3). Hence, it can be inferred that CBC may exert its modulatory effects on the transcription of these genes through the downregulation of other transcription factors such as STAT-3, CCAAT/enhancer binding protein (C/EBP), cAMP response element-binding protein (CREB), and activator protein-1 (AP-1). 

Consistently, it has been demonstrated that the treatment of Caco-2 cells with cAMP, known to induce IL-6 production, leads to an increase in the DNA binding activity of the transcription factors CREB, C/EBP, and AP-1. However, there is no notable effect on the activity of NF-κB [51]. Interestingly, CBC has been demonstrated to induce the phosphorylation of ERK and suppress forskolin-induced cAMP stimulation [49].

Our findings suggest that the anti-inflammatory effects of CBC are mediated by modulating the transcription of cytokines and COX-2. This modulation likely involves a pathway independent of NF-κB.

In contrast to NF-κB, CBC downregulated the IL-6/TYK-2/STAT-3 axis (Figure 3, Figure 4 and Figure 5), suggesting the possible modulatory effects of CBC on the cytokine storm.

IL-6 preferentially activates gene expression dependent on STAT-3 [52,53]. The activation of STAT-3 by IL-6 plays a crucial role in the pathogenesis of inflammation-induced diseases [54]. Furthermore, IL-6 is known to activate COX-2 through its trans-signaling pathway, and the IL-6-STAT-3-COX-2 pathway has been shown to be important in inflammation-induced malignancies and is controlled by positive feedback regulation [55,56]. It is interesting to note that CBC has been shown to downregulate the activation of the IL-6/TYK-2/STAT-3 pathway, suggesting a possible role for this pathway in the modulatory effects of CBC on COX-2 transcription. 

Existing evidence indicates that IL-1β, similar to IL-6, has the ability to stimulate the activation of STAT-3, thus presenting an additional mechanism through which CBC may exert its inhibitory effects on STAT-3 activation [57].

According to our results, CBC exerts regulatory effects on the mature form of IL-1β by inhibiting both phases of NLRP3 inflammasome activation. Similar to THCV, the downregulation of NLRP3, PANX-1, and P2X7 at the mRNA and/or protein levels plays a significant role in mediating the inhibitory effects of CBC on the second phase of NLRP3 inflammasome activation (Figure 7, Figure 8 and Figure 9).

### 3.3. The Effect of CBN on LA/LPS Induced Macrophages 

CBN also exhibited downregulatory effects on proinflammatory cytokines and COX-2 at the protein level. Both doses of CBN inhibited the levels of IL-1β, TNFα, IL-6, and COX-2 proteins in the LPS/LA-induced macrophages, indicating the anti-inflammatory properties of CBN (Figure 1 and Figure 2). Consistent with our observations regarding CBN’s ability to alleviate the mature form of TNFα, Lace et al. documented a reduction in TNFα levels in human primary leukocytes with 10 μM CBN [58]. 

Furthermore, both doses of CBN demonstrated modulatory impacts on the activation of NFκB, although the inhibitory effects were more pronounced with the lower dose (Figure 3). The ratio of P-NFκB to total NFκB suggests that CBN may primarily exert its effects on the activation of NFκB through the downregulation of phosphorylation rather than transcriptional regulation.

Despite the modulatory effects of CBN on the protein levels of COX-2 and Pro-TNFα, it upregulated the transcription of these genes, suggesting a potential post-transcriptional mode of action for CBN in mitigating the COX-2 and Pro-TNFα proteins (Figure 9).

Regarding the response of the IL-1β and IL-6 transcripts, 15 μM CBN exhibited inhibitory effects (Figure 9), likely mediated through the downregulation of NF-κB phosphorylation and activation. However, the lower dose did not significantly affect their expression.

To assess the effects of CBN on the second phase of NLRP3 inflammasome activation and the subsequent cleavage of Pro-IL-1β, the protein levels of NLRP3 and Pro-caspase-1 were examined. CBN demonstrated modulatory effects on the protein levels of NLRP3 in the LA-induced macrophages (Figure 7), suggesting the potential mitigative effects of CBN on the second step of NLRP3 inflammasome activation.

Interestingly, while neither dose of CBN affected the response of PANX-1 (55 kDa) at the protein level, both doses inhibited the cleavage and activation of PANX-1. This implies that the inhibitory effects of CBN on the second step of NLRP3 inflammasome activation may be mediated, at least partially, through the modulation of PANX-1 cleavage, leading to the opening of PANX-1 and subsequent release of extracellular ATP.

Additionally, the qRT-PCR results for PANX-1 provide further evidence suggesting that CBN prevents the opening of the PANX-1 transporter by downregulating its cleavage rather than affecting its transcription.

Both doses of CBN demonstrated the downregulation of STAT-3 activation (Figure 4c). However, unlike THCV and CBC, this effect does not appear to be mediated through the mitigation of P-TYK-2. Instead, it is likely mediated through the modulation of other JAKs.

As previously mentioned, like IL-6, IL-1β plays an important role in the activation of STAT-3. In line with this, Liang et al. (2020) showed that IL-1β exerts its stimulatory effects on the activation of STAT-3 through the downregulation of Kruppel-like factor 2 (KLF2), which, in turn, leads to the stimulation of heat shock protein H1 (HSPH1). HSPH could interact with STAT-3 and enhance its phosphorylation. These findings suggest that activation of the IL-1β/KLF2/HSPH1 pathway facilitates STAT-3 phosphorylation [57].

The downregulatory effects of CBN on mature IL-1β suggest that CBN may exert its mitigative effects on STAT-3 activation through the IL-1β/KLF2/HSPH1 pathway.

Furthermore, it was discovered that CBN also has the ability to reduce the increased amount of P-STAT-1 in macrophages stimulated with LPS (Figure 6). This finding suggests that CBN likely inhibits the production of IL-6 by decreasing the levels of ARID5A and subsequently affecting the stability of IL-6 mRNA. 

To confirm the results of Western blotting with the mature form of IL-1β, ELISA with the collected supernatant from cultured cells was performed. The ELISA results for CBC and THCV showed that both doses decreased the secretion of IL-1β in the medium in a dose-dependent manner, which was in accordance with the results of Western blotting with the mature form of IL-1β. Interestingly, both doses of CBN increased the secretion of IL-1β in the medium in a dose-dependent manner. This was in contrast to the response of the mature form of IL-1β to both doses of CBN. 

Phospholipase A2 (PLA2) is an enzyme that releases arachidonic acid (AA) from membrane phospholipids. AA serves as a substrate for COX enzymes including COX-1 and COX-2, leading to the production of prostaglandins (PGs). Specifically, cytosolic group IV PLA2 (cPLA2) acts on AA, providing it as a substrate for COX-2 and resulting in the synthesis of PGE2, a type of prostaglandin. Notably, COX-2 is induced by proinflammatory cytokines and LPS and is involved in mediating inflammatory responses [59]. Based on our qRT-PCR findings, it was observed that both THCV and CBC decreased the expression of cPLA2 in macrophages stimulated with LPS. This suggests that these two phytocannabinoids have an additional target to exert their inhibitory effects on inflammation (Figure 9). 

ADAR1, an enzyme known as adenosine deaminase acting on RNA, is involved in the conversion of adenosine to inosine in both precursor and mature mRNA molecules. This process of RNA editing mediated by ADAR1 contributes to the generation of diverse mRNA and protein isoforms. ADAR1 exhibits widespread expression across various cell types, indicating its ubiquitous presence [60]. 

Despite numerous studies conducted by various research groups on the role of ADAR1 in inflammation, its precise function remains a subject of controversy. Some studies have proposed an anti-inflammatory role for ADAR1 [61,62]. However, most published studies consistently indicate that ADAR1 enhances the inflammatory response [60,63,64].

The ADAR1 p150 isoform plays a role in alveolar macrophages (AMs) in the progression of lung inflammation through modulating the release of inflammatory mediators, specifically the chemokine MIP-1 and the anti-inflammatory cytokine IL-10 [60]. 

In relation to the crucial role of ADAR-1 in inflammation, a recent study conducted by Cai et al. (2023) demonstrated that the targeted knockout of ADAR-1 specifically in macrophages in mice led to a notable decrease in the expression levels of key inflammatory markers such as IL-1β, IL-6, TNFα, and iNOS [65].

In our study, we made a notable discovery that both THCV and CBC, at different doses, suppressed the transcriptional activity of ADAR-1 in macrophages induced by LPS. This finding suggests the existence of a potential post-transcriptional mechanism in the form of RNA editing through which these two cannabinoids exert their anti-inflammatory effects (Figure 9).

## 4. Materials and Methods

### 4.1. Cell Culture and Treatments 

THP-1 monocytes (ATCC TIB-202) were sourced from ATCC. The cells were grown in cell culture dishes at a concentration of 2–4 × 10^5^ viable cells/mL using RPMI 1640 (350-000-CL) medium, supplemented with 10% fetal bovine serum (FBS) (Cat: 97068-085) from Avantor, Radnor, PA, USA, and 5% antibiotic cocktails (Cat: LS15140122) from Gibco, Life Technologies Inc., Burlington, ON, Canada. Cell passages 8–13 were used for all experiments. To differentiate monocytes into macrophages, non-adherent monocytes were treated for 48 h with 50 ng of phorbol 12-myristate 13-acetate (PMA) (Enzo Life Sciences, Toronto, ON, Canada, Cat: BML-PE160-0005). Adherent macrophages were then washed with PBS (1X) twice and allowed to recover for 24 h in PMA-free medium. 

Macrophages were treated with two different concentrations of three cannabinoids: tetrahydrocannabivarin (THCV) (T-094-1ML), cannabichromene (CBC) (20675-51-8), and cannabinol (CBN) (521-35-7), all from Sigma Aldrich, Oakville, ON, Canada, or an equivalent volume of vehicle control, for 1 h. Subsequently, the cells were stimulated with 500 ng/mL lipopolysaccharide (LPS, from Escherichia coli O111:B4) (L4391, Sigma Aldrich, Oakville, ON, Canada) for 4 h to induce pro inflammatory responses. To activate the second phase of NLRP3 inflammasome, the LPS-induced cells were incubated with 5 mM adenosine 5′-triphosphate (ATP) disodium salt hydrate (A6419, Sigma Aldrich, Canada) for 30 min.

In this research, our objective was to examine the anti-inflammatory effects of three minor phytocannabinoids, utilizing two different doses—a targeted optimal dose and a higher dose. Through our optimization trials and in consideration of prior research involving major cannabinoids [13,31], we established 5 μM as the optimal concentration. For the high dose, we selected 15 μM for THCV and CBN, as this concentration demonstrated the highest effectiveness without causing noteworthy cytotoxicity. In the case of CBC, its cytotoxic effects at 15 μM compelled us to utilize both the optimal dosage and a lower dosage (2.5 μM).

After the experimental treatments, RNA and protein were extracted from the treated cells for quantitative real-time polymerase chain reaction (qRT-PCR) and Western blot analysis, respectively. The supernatant from the cell culture was collected for further analysis through ELISA. 

### 4.2. Western Blotting 

Following the collection of the supernatant for the ELISA assay, the cells were washed twice with cold PBS. Subsequently, the cells were lysed in RIPA buffer and centrifuged at 13,000 rpm for 15 min. The resulting supernatant was transferred to a new microtube, and the protein content was quantified using the Bradford assay. The lysates were then used for Western blotting.

To perform the Western blot, the protein samples were resolved on polyacrylamide gels in concentrations of 8%, 10%, 12%, and 15%. The separated proteins were then transferred onto polyvinylidene difluoride (PVDF) (Amersham Hybond^®^ P, GE Healthcare) membranes for further analysis. The membranes were blocked using PBST solution containing 5% milk and incubated with the primary antibody overnight at 4 °C. Subsequently, the membranes were washed three times with PBST.

Antibody-linked peroxidase was employed to detect immunoreactivity, and this was made visible using the ECL Plus Western Blotting Detection System (GE Healthcare, Chicago, IL, USA). The intensity of the bands was quantified and adjusted based on the intensity of house-keeping proteins using ImageJ.

The information on the primary antibodies is listed in Appendix A.

### 4.3. qRT-PCR

Following cellular RNA extraction using the TRIzol^TM^ reagent (15596026) as per the manufacturer’s instructions, the total RNA was quantified, and a portion of it was utilized for cDNA synthesis. For cDNA synthesis, 1 μg of total RNA was used (cDNA synthesis kit, Cat: 1708840). Subsequently, 1 μL of the synthesized cDNA was used as a template for performing q-PCR. The q-PCR reactions were conducted using SsoAdvanced^TM^ Universal Inhibitor-Tolerant SYBR Green Supermix (Cat: 172-5016). The primers for q-PCR analysis were designed using online IDT software (Appendix A).

### 4.4. Enzyme-Linked Immunosorbent Assay (ELISA)

To examine the impact of the treatments on the release of IL-β, we conducted an ELISA assay using the Human IL-1β /IL-1F2 Quantikine ELISA Kit (Catalog #: DLB50). In each well, 200 μL of the standard, control, or sample was added, followed by a 2 h incubation at room temperature. Subsequently, the wells were washed three times with washing buffer, and then 200 μL of human IL-1β conjugate was added to each well, which was covered with a new adhesive strip. After 1 h of incubation at room temperature, another round of aspiration/wash was performed. Afterward, 200 μL of Substrate Solution was added to each well, and the plate was incubated for 20 min at room temperature. Finally, 50 μL of Stop Solution was added to each well, and the optical density of each well was measured at 450 nm and 570 nm using a specific device (SpectraMax i3x Multi-Mode Microplate Reader, Molecular Devices, San Jose, CA, USA). To account for optical imperfections in the plate, the readings at 570 nm were subtracted from the readings at 450 nm. It is important to note that due to the high concentration of secreted IL-β in the medium, the samples had to be diluted by a factor of 50 before running the ELISA assay.

### 4.5. MTT Assay

An MTT assay was utilized to assess the influence of THCV, CBC, and CBN phytocannabinoids on the cell viability of THP-1 macrophages induced by LPS + ATP. THP-1 monocyte cells were seeded into individual wells of a 96-well plate at a density of 1 × 10^6^ cells per well, followed by incubating with PMA to differentiate into macrophages. Macrophages were incubated in 100 μL of culture medium containing either 5 μM of each phytocannabinoid or an equivalent volume of vehicle (methanol) for a duration of 1 h. Subsequently, the cells were incubated with medium containing the phytocannabinoids, vehicle, and/or LPS + ATP for 24 h at +37 °C and 5% CO_2_. After the incubation period, 10 μL (equivalent to 10% of the well’s content) of the labeling reagent 3-(4,5-dimethylthiazol-2-yl)-2,5-diphenyltetrazolium bromide (MTT) from Roche Diagnostics GmbH was introduced into each well. The microplate was then placed in a humidified environment (+37 °C, 5% CO_2_) for an extra 4 h. Afterward, 100 μL of solubilization solution (SDS 10% in diluted hydrochloric acid) was added to each well and the plate was left to stand overnight in a humidified incubator (+37 °C, 5% CO_2_). The absorbance at 595 nm was then measured using a plate reader (FLUOstar Omega, BMG LABTECH, Offenburg, Germany).

### 4.6. Statistics

The collected data were subjected to statistical analysis using one-way analysis of variance (ANOVA) followed by Dunnett’s test to compare the mean values, performed using GraphPad Prism 6 software.

## 5. Conclusions

In summary, THCV and CBN demonstrated the inhibition of P-NF-κB, indicating a mechanism through which these two phytocannabinoids downregulate the transcription of proinflammatory genes. Additionally, THCV can suppress the PANX-1/PANX1 cleavage/P2X7 pathway, leading to the reduction in NLRP3 inflammasome assembly, Pro-Caspase-1 activation, and IL-1β production. It also inhibits the NFκB/IL-6/TYK-2/STAT-3 and TYK-2/STAT-1 pathways, indicating its potential in combating cytokine storms.

Similarly, CBN inhibits NLRP3 inflammasome assembly by mitigating PANX-1 and PANX1 cleavage. It also exhibits inhibitory effects on STAT-3 and STAT-1, although the mechanism may differ from TYK-2 downregulation.

Furthermore, CBC’s ability to mitigate NLRP3 inflammasome assembly, Pro-Caspase-1 activation, and IL-1β production is linked to the modulation of the PANX-1/P2X7 axis in LPS-induced macrophages. Although CBC does not affect NF-κB activation, it does show potential in suppressing the IL-6/TYK-2/STAT-3 pathway, thus potentially alleviating cytokine storms in LPS-induced macrophages.

Finally, both CBC and THCV negatively affect ADAR1, suggesting a possible involvement of RNA editing in their inhibitory effects on pro-inflammatory cytokines and proteins (Figure 12).

## Figures and Tables

**Figure 1 molecules-28-06487-f001:**
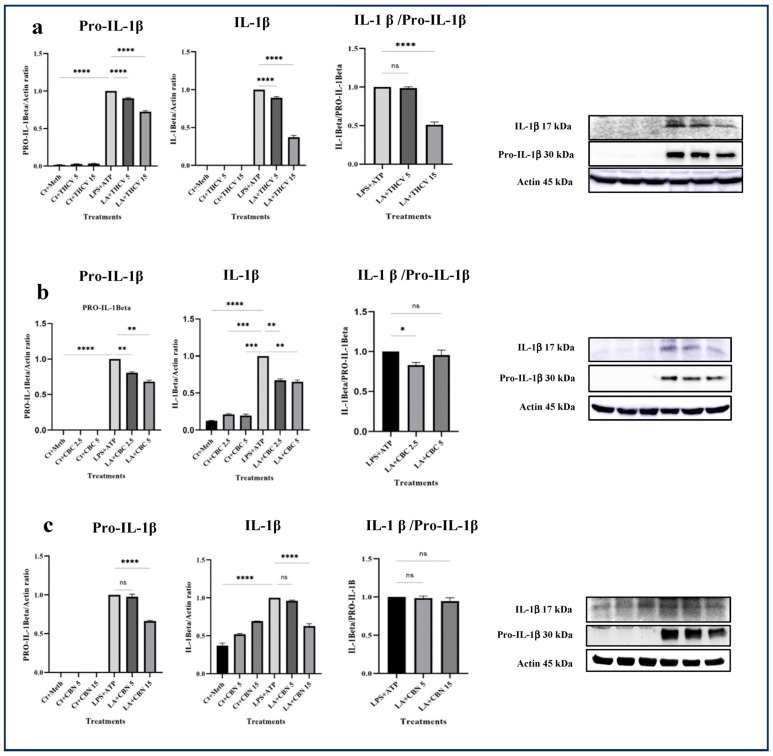
THCV, CBC, and CBN mitigated the production of both the immature and mature forms of IL-1β in LA-induced macrophages. (**a**) Western blot analysis of Pro-IL-1β, IL-1β, and IL-1β/Pro-IL-1β in THCV-pretreated LA-induced macrophages, with β-Actin used as a loading control. (**b**) Western blot analysis of Pro-IL-1β, IL-1β, and IL-1β/Pro-IL-1 in CBC-pretreated LA-induced macrophages, with β-Actin used as a loading control. (**c**) Western blot analysis of Pro-IL-1β, IL-1β, and IL-1β/Pro-IL-1 in the CBN-pretreated LA-induced macrophages, with β-Actin used as a loading control. Relative densitometry was measured using ImageJ. Data are presented as the mean value +/− SD. Abbreviations: LA: LPS + ATP, Ct: Control, Meth: Methanol. Panels on the right show representative blot images. The asterisks show significant difference, where one—*p* < 0.05, two—*p* < 0.01, three—*p* < 0.001, four—*p* < 0.0001; ns—non-significant.

**Figure 2 molecules-28-06487-f002:**
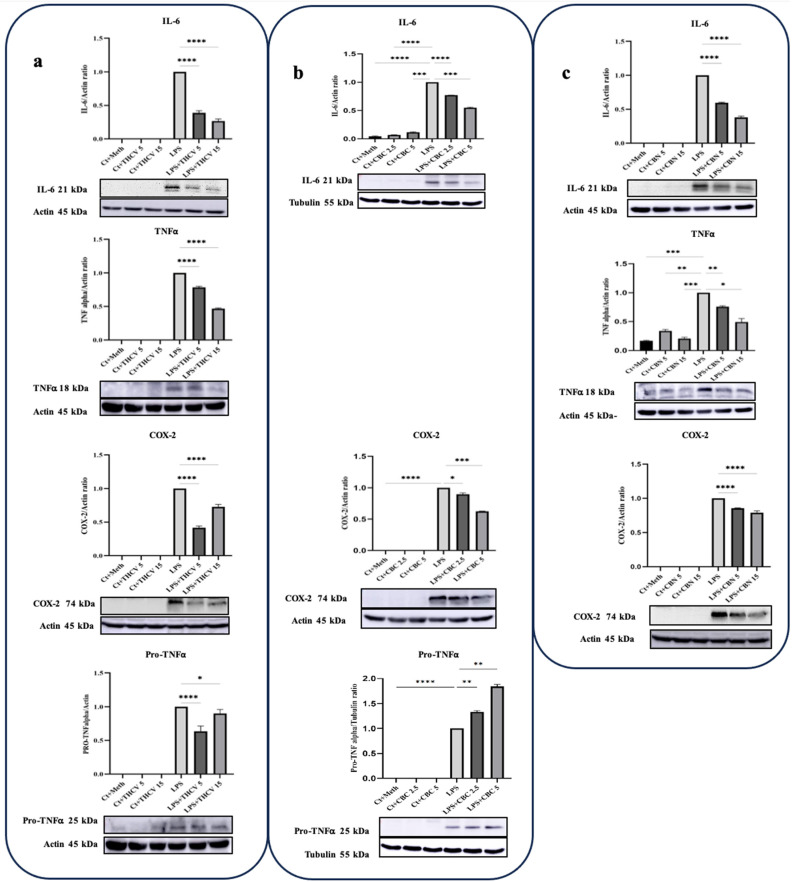
The western blot results showing the effects of two doses of THCV, CBC, and CBN on IL-6, TNFα, COX-2, and Pro-TNFα proteins in the LPS-induced macrophages. (**a**) Western blot analysis of IL-6, TNFα, COX-2, and Pro-TNFα in the THCV-pretreated LPS-induced macrophages, with β-Actin used as a loading control. (**b**) Western blot analysis of IL-6 and Pro-TNFα with α-Tubulin used a loading control, and COX-2 with β-Actin used as a loading control in CBC-pretreated LPS-induced macrophages, (**c**) Western blot analysis of IL-6, TNFα and COX-2 in CBN-pretreated LPS-induced macrophages, with β-Actin used as a loading control. Relative densitometry was measured using ImageJ. All data are presented as the mean value +/− SD. Abbreviations: LA: LPS + ATP, Ct: Control, Meth: Methanol. Panels under the graphics show representative blot images. The asterisks show significant difference, where one—*p* < 0.05, two—*p* < 0.01, three—*p* < 0.001; four—*p* < 0.0001.

**Figure 3 molecules-28-06487-f003:**
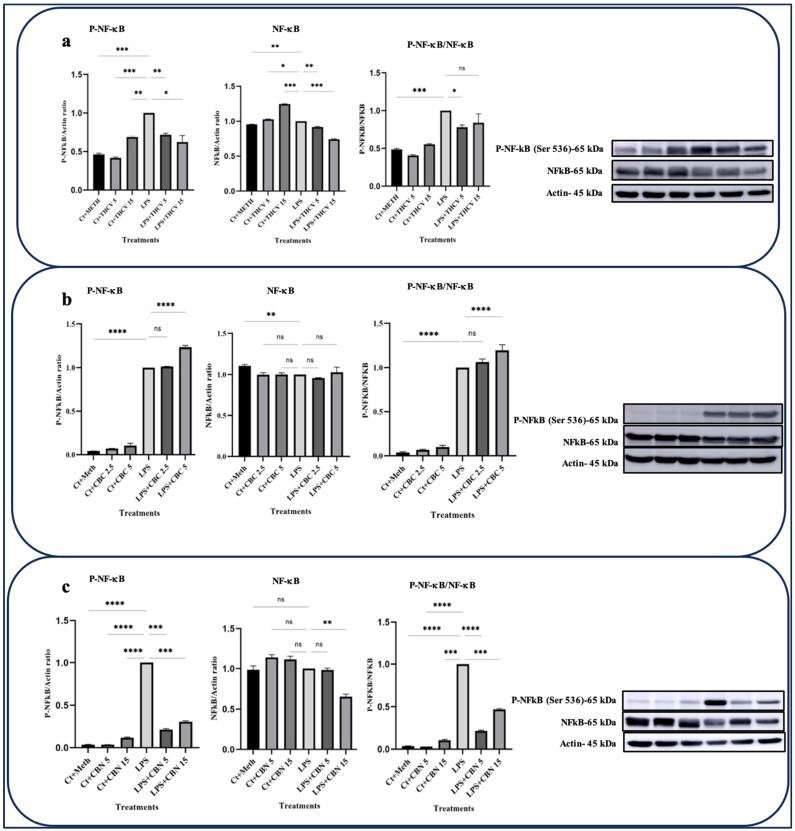
THCV and CBN mitigate the activation of NF-κB in the LPS-induced macrophages. (**a**) Western blot analysis of P-NF-κB, NF-κB, and P-NF-κB/NF-κB in the THCV-pretreated LPS-induced macrophages with β-Actin used as a loading control. (**b**) Western blot analysis of P-NF-κB, NF-κB, and P-NF-κB/NF-κB in the CBC-pretreated LA-induced macrophages with β-Actin used as a loading control. (**c**) Western blot analysis of P-NF-κB, NF-κB, and P-NF-κB/NF-κB in the CBN-pretreated LPS-induced macrophages, with β-Actin used as a loading control. Relative densitometry was measured using ImageJ. All data are presented as the mean value +/− SD. Abbreviations: LA: LPS + ATP, Ct: Control, Meth: Methanol. Panels on the right show representative blot images. The asterisks show significant difference, where one—*p* < 0.05, two—*p* < 0.01, three—*p* < 0.001; four—*p* < 0.0001; ns—non-significant.

**Figure 4 molecules-28-06487-f004:**
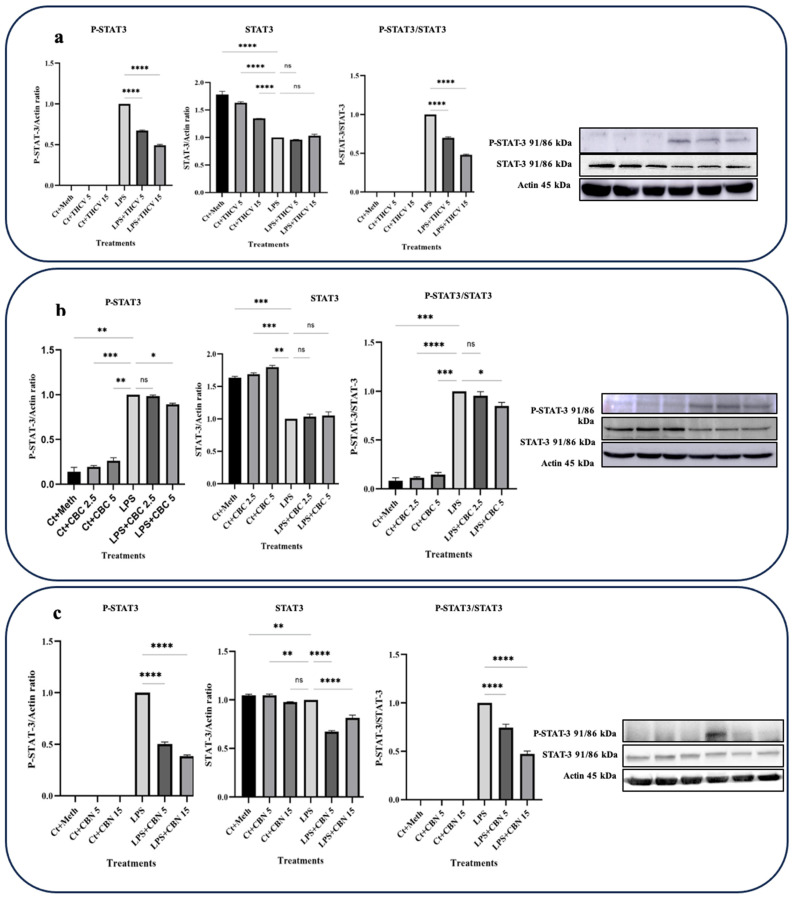
THCV, CBC, and CBN mitigate the phosphorylation of STAT3 in LPS-induced macrophages. (**a**) Western blot analysis of phospho-STAT3, total-STAT3, and P-STAT3/t-STAT3 in the THCV-pretreated LPS-induced macrophages with β-Actin used as a loading control. (**b**) Western blot analysis of phospho-STAT3, total-STAT3, and P-STAT3/t-STAT3 in the CBC-pretreated LA-induced macrophages with β-Actin used as a loading control. (**c**) Western blot analysis of phospho-STAT3, total-STAT3, and P-STAT3/t-STAT3 in the CBN-pretreated LPS-induced macrophages with β-Actin used as a loading control. Relative densitometry was measured using ImageJ. All data are presented as the mean value +/− SD. Abbreviations: LA: LPS + ATP, Ct: Control, Meth: Methanol. Panels on the right show representative blot images. The asterisks show significant difference, where one—*p* < 0.05, two—*p* < 0.01, three—*p* < 0.001; four—*p* < 0.0001; ns—non-significant.

**Figure 5 molecules-28-06487-f005:**
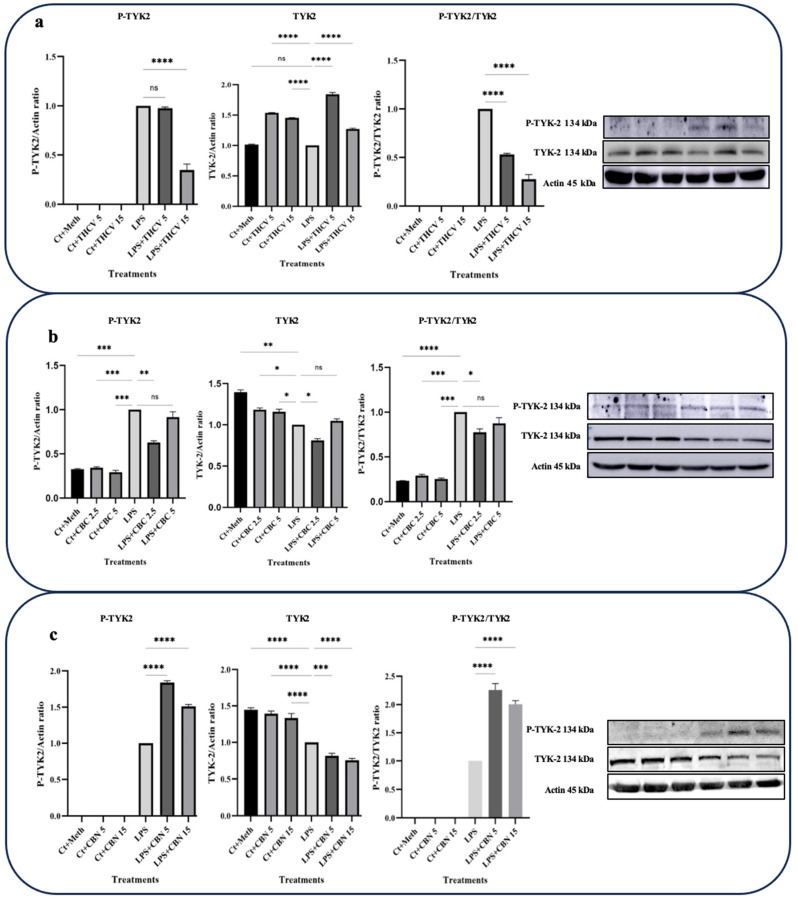
THCV and CBC show a mitigative impact on the activation of TYK-2 in the LPS-induced macrophages. (**a**) Western blot analysis of P-TYK-2, total-TYK-2, and P-TYK-2/t-TYK-2 in the THCV-pretreated LPS-induced macrophages with β-Actin used as a loading control. (**b**) Western blot analysis of P-TYK-2, total-TYK-2, and P-TYK-2/t-TYK-2 in CBC-pretreated LA-induced macrophages, with β-Actin used as a loading control. (**c**) Western blot analysis of P-TYK-2, total-TYK-2 and P-TYK-2/t-TYK-2 in CBN-pretreated LPS-induced macrophages, with β-Actin used as a loading control. Relative densitometry was measured using ImageJ. All data are presented as the mean value +/− SD. Abbreviation: LA: LPS + ATP, Ct: Control, Meth: Methanol, PANX-1: pannexin-1. Panels on the right show representative blot images. The asterisks show significant difference, where one—*p* < 0.05, two—*p* < 0.01, three—*p* < 0.001; four—*p* < 0.0001; ns—non-significant.

**Figure 6 molecules-28-06487-f006:**
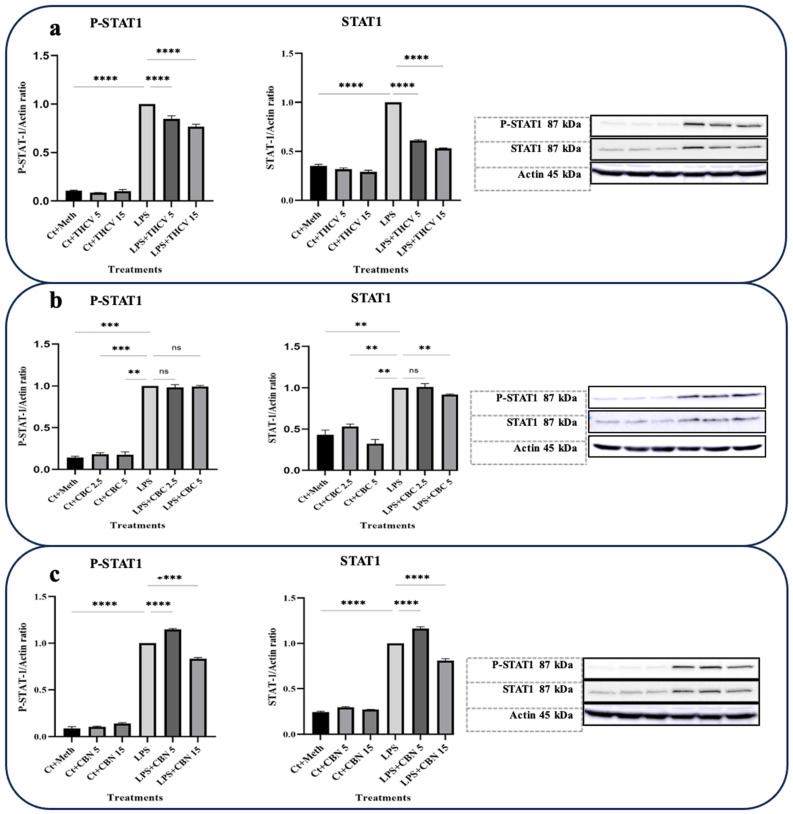
THCV and CBN mitigate the phosphorylation of STAT1 in the LPS-induced macrophages. (**a**) Western blot analysis of phospho-STAT1, total-STAT1, and P-STAT1/t-STAT1 in the THCV-pretreated LPS-induced macrophages with β-Actin used as a loading control. (**b**) Western blot analysis of phospho-STAT1, total-STAT1, and P-STAT1/t-STAT1 in the CBC-pretreated LA-induced macrophages with β-Actin used as a loading control. (**c**) Western blot analysis of phospho-STAT1, total-STAT1, and P-STAT1/t-STAT1 in the CBN-pretreated LPS-induced macrophages with β-Actin used as a loading control. Relative densitometry was measured using ImageJ. All data are presented as the mean value +/− SD. *N* = 3 measurements. Abbreviations: Ct: Control, Meth: Methanol. Panels show representative blot images. The asterisks show significant difference, where two—*p* < 0.01, three—*p* < 0.001; four—*p* < 0.0001; ns—non-significant.

**Figure 7 molecules-28-06487-f007:**
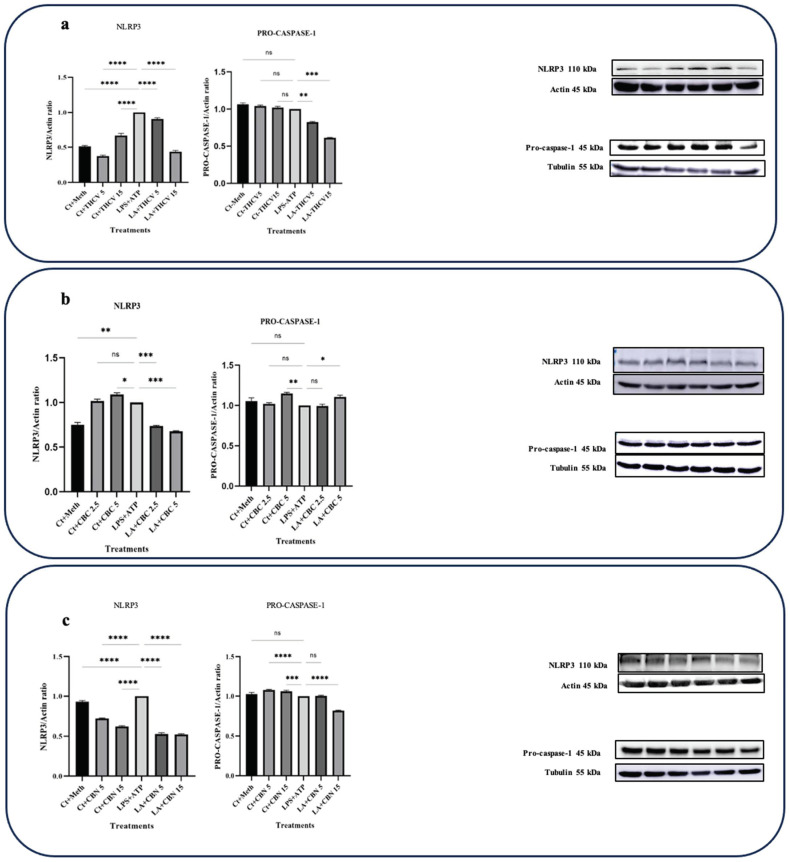
THCV, CBC, and CBN mitigate the level of the NLRP3 protein in LA-induced macrophages. (**a**) Western blot analysis of NLRP3 and Pro-caspase-1 in the THCV-pretreated LPS-induced macrophages with β-Actin and α-Tubulin used as the loading controls. (**b**) Western blot analysis of NLRP3 and Pro-caspase-1 in the CBC-pretreated LA-induced macrophages with β-Actin and α-Tubulin used as the loading controls. (**c**) Western blot analysis of NLRP3 and Pro-caspase-1 in the CBN-pretreated LPS-induced macrophages with β-Actin and α-Tubulin used as the loading controls. Relative densitometry was measured using ImageJ. All data are presented as the mean value +/− SD. Abbreviations: LA: LPS + ATP, Ct: Control, Meth: Methanol. Panels on the right show representative blot images. The asterisks show significant difference, where one—*p* < 0.05, two—*p* < 0.01, three—*p* < 0.001; four—*p* < 0.0001; ns—non-significant.

**Figure 8 molecules-28-06487-f008:**
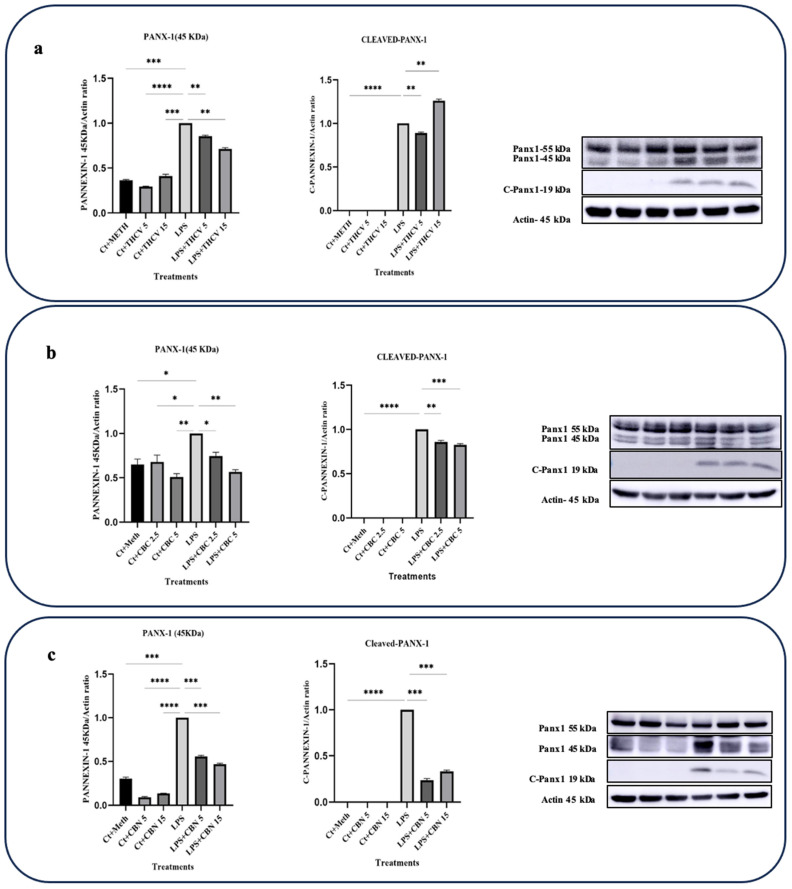
THCV, CBC, and CBN mitigate the level of PANX-1 (45 KDa) and Cleaved-PANX-1 in the LPS-induced macrophages. (**a**) Western blot analysis of PANX-1 (45 KDa) and Cleaved-PANX-1 in the THCV-pretreated LPS-induced macrophages with β-Actin used as a loading control. (**b**) Western blot analysis of PANX-1 (45 KDa) and Cleaved-PANX-1 in CBC-pretreated LA-induced macrophages with β-Actin used as a loading control. (**c**) Western blot analysis of PANX-1 (45 KDa) and Cleaved-PANX-1 in the CBN-pretreated LPS-induced macrophages with β-Actin used as a loading control. Relative densitometry was measured using ImageJ. All data are presented as the mean value +/− SD. Abbreviations: LA: LPS + ATP, Ct: Control, Meth: Methanol, PANX-1: pannexin-1. Panels on the right show representative blot images. The asterisks show significant difference, where one—*p* < 0.05, two—*p* < 0.01, three—*p* < 0.001; four—*p* < 0.0001.

**Figure 9 molecules-28-06487-f009:**
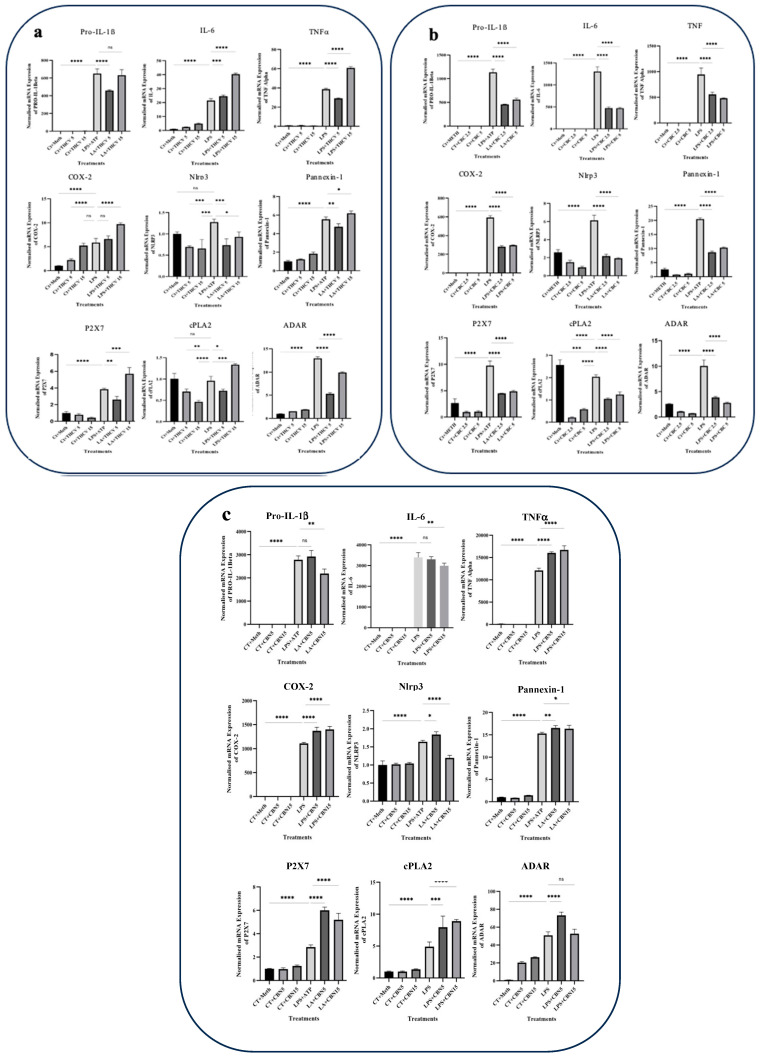
The quantitative real-time polymerase chain reaction (qRT-PCR) findings for the mRNA expression levels of *IL-1β*, *IL-6*, *TNFα*, *COX-2*, *Nlrp3*, *Pannexin-1*, *P2X7*, *cPLA2*, and *ADAR-1* in response to THCV, CBC, and CBN in the LPS-induced macrophages. (**a**) The normalized mRNA expression levels of *IL-1β*, *IL-6*, *TNFα*, *COX-2*, *Nlrp3*, *PANX-1*, *P2X7*, *cPLA2*, and *ADAR-1* in response to two doses of THCV. (**b**) The normalized mRNA expression levels of *IL-1β*, *IL-6*, *TNFα*, *COX-2*, *Nlrp3*, *PANX-1*, *P2X7*, *cPLA2*, and *ADAR-1* in response to two doses of CBC. (**c**) The normalized mRNA expression levels of *IL-1β*, *IL-6*, *TNFα*, *COX-2*, *Nlrp3*, *PANX-1*, *P2X7*, *cPLA2*, and *ADAR-1* were analyzed in response to two doses of CBN. All data are presented as the mean value +/− SD. Abbreviations: LA: LPS + ATP, Ct: Control, Meth: Methanol. Panels at the bottom show representative blot images. The asterisks show significant difference, where one—*p* < 0.05, two—*p* < 0.01, three—*p* < 0.001; four—*p* < 0.0001; ns—non-significant.

**Figure 10 molecules-28-06487-f010:**
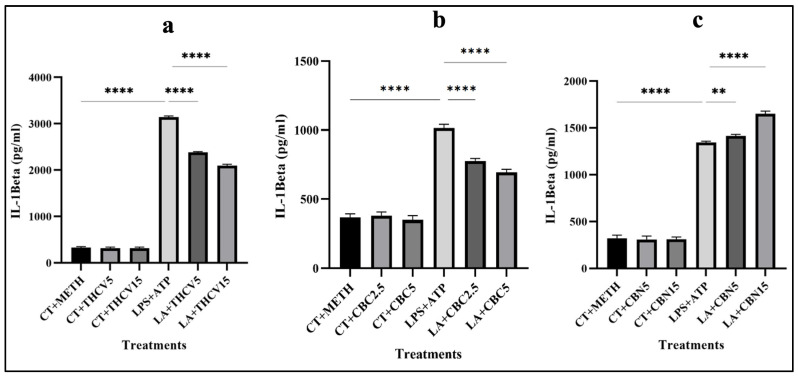
THCV and CBC mitigate the secretion of mature IL-1β in LA-induced macrophages. (**a**) The effects of 5 μM and 15 μM THCV on secreted IL-1β. (**b**) The effects of 2.5 μM and 5 μM CBC on secreted IL-1β. (**c**) The effects of 5 μM and 15 μM CBN on secreted IL-1β. All data are presented as the mean value +/− SD. Abbreviations: LA: LPS + ATP, Ct: Control, Meth: Methanol. Panels at the bottom show representative blot images. The asterisks show significant difference, where two—*p* < 0.01, four—*p* < 0.0001.

**Figure 11 molecules-28-06487-f011:**
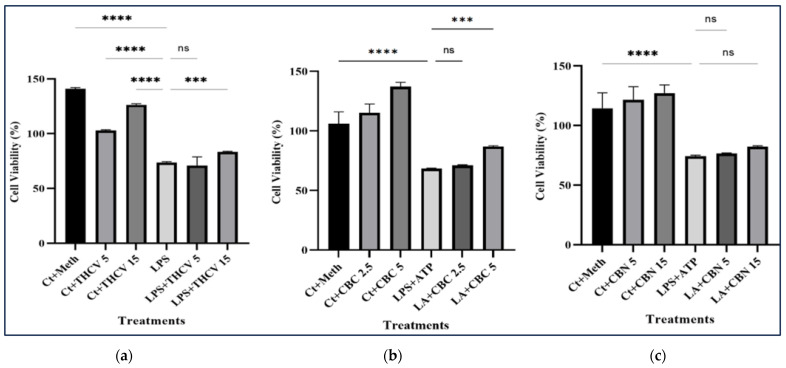
The effects of two doses of THCV, CBC, and CBN on the viability of the LPS + ATP-induced macrophages. (**a**) The effects of 5 μM and 15 μM THCV on the viability of the LPS + ATP-induced macrophages. (**b**) The effects of 2.5 μM and 5 μM CBC on the viability of the LPS + ATP-induced macrophages. (**c**) The effects of 5 μM and 15 μM CBN on the viability of the LPS + ATP-induced macrophages. All data are presented as the mean value +/− SD. Abbreviations: LA: LPS + ATP, Ct: Control, Meth: Methanol. Panels at the bottom show representative blot images. The asterisks show significant difference, where three—*p* < 0.001; four—*p* < 0.0001; ns—non-significant.

**Figure 12 molecules-28-06487-f012:**
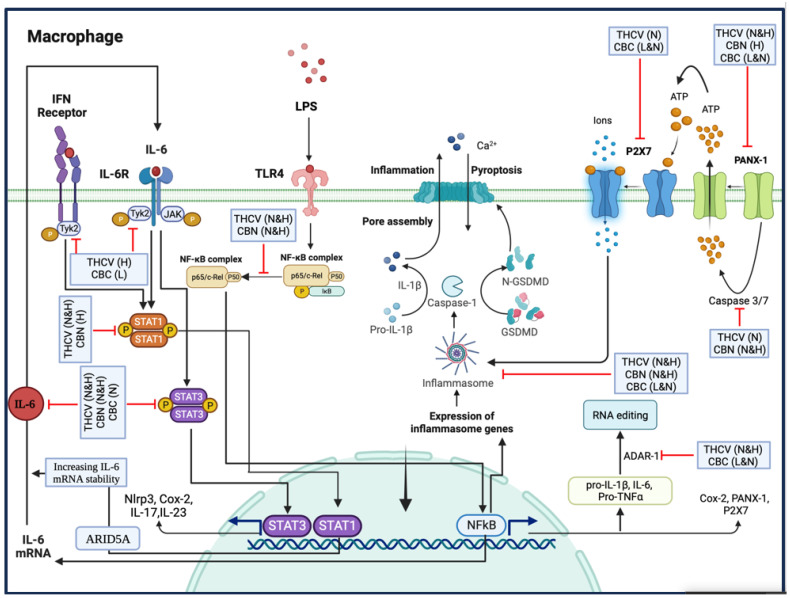
The figure presents the main findings of the study, highlighting the effects of THCV, CBC, and CBN on various molecular pathways involved in inflammation. THCV, CBN inhibit P-NF-κB, which may contribute to their anti-inflammatory properties. Furthermore, THCV demonstrates the ability to mitigate the PANX-1/ P2X7 axis, leading to the suppression of NLRP3 inflammasome assembly, Pro-caspase-1 activation, and IL-1β production. THCV also exhibits inhibitory effects on the NF-κB/IL-6/TYK2/STAT3 pathways, indicating its potential to combat cytokine storms. CBN also inhibits the assembly of the NLRP3 inflammasome partly by suppressing PANX-1 and PANX1 opening. Similar to THCV, CBN also shows inhibitory effects on STAT3 and STAT1, although the mechanism is likely different from TYK2 downregulation. In addition, the modulation of the PANX-1/P2X7 axis plays a crucial role in the mitigating effects of CBC on NLRP3 inflammasome assembly, Pro-Caspase-1 activation, and IL-1β production. Although CBC does not downregulate NF-κB activation, it does mitigate STAT3, suggesting its potential in suppressing cytokine storms in LPS-induced macrophages. It is important to note that ADAR1 responds negatively to both CBC and THCV, suggesting a possible involvement of RNA editing modulation in the inhibitory effects of CBC and THCV on pro-inflammatory cytokines and proteins. Abbreviations: L: (Low) 2.5 µM, N: (Normal) 5 µM, H: (High) 15 µM.

## Data Availability

Raw Western blot images were submitted to Molecules.

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
