# Peer review of "Anti-Inflammatory Effects of Minor Cannabinoids CBC, THCV, and CBN in Human Macrophages"

_molecules, 2023, doi:10.3390/molecules28186487_

Round 1

Reviewer 1 Report

The manuscript by Igor Kovalchuk and co-workers report a study about the anti-inflammatory effects of three significant minor phytocannabinoids: CBC, THCV, and CBN. The authors assayed different doses of phytocannabinoids or vehicle on THP-1 macrophages stimulated with  LPS and ATP to elucidate the mechanism of action of anti-inflammatory phytochemicals.

 The manuscript shows important information and was presented in a well structured way. I would only recommend the following to the authors:

It is advisable to report how the phytocannabinoids were prepared and manipulated for their administration in the biological trials.

What was the maximum concentration of vehicle used (methanol)? And is convenient to report the effect of the vehicle in trials.

Is methanol the most convenient vehicle to use in this type of biological assay?

Finally, it is recommended to improve the quality of the images.

Good luck

Author Response

Q: It is advisable to report how the phytocannabinoids were prepared and manipulated for their administration in the biological trials.

A: We did not prepare the phytocannabinoids. All three phytocannabinoids utilized in the research were obtained from Millipore Sigma (Canada). They were supplied in a methanol solution at a concentration of 1 mg/ml by manufacturer. The respective catalog numbers for each phytocannabinoid are as follows:

Tetrahydrocannabivarin (THCV) (T-094-1ML),

Cannabichromene (CBC) (20675-51-8)

Cannabinol (CBN) (521-35-7).

Q: What was the maximum concentration of vehicle used (methanol)? And is convenient to report the effect of the vehicle in trials.

A: The methanol quantity used in our experiments matched the highest volume of the utilized phytocannabinoids. For instance, in the case of treating with 15 μM THCV, we employed 52.37 μl of methanol within a 10 ml medium-filled plate.

We did not aim to report the effect of vehicle in our trials.

Q: Is methanol the most convenient vehicle to use in this type of biological assay?

A: We obtained the phytocannabinoids as a methanol solution and utilized them in the same solvent. Methanol, while not optimal due to potential adverse effects, isn't the ideal choice for biological assays, especially in vivo studies.

It is noteworthy that we evaluated methanol's cytotoxicity using the MTT assay and found that it did not significantly compromise cell viability compared to the LPS+ATP group.

Q: Finally, it is recommended to improve the quality of the images.

A: We improved the quality of all figures.

Reviewer 2 Report

Page 1; Abstract; The authors should follow the abstract's style of the journal, which is not a structured one. The authors are recommended to add a conclusion statement at the end of the abstract to highlight the significance of the obtained results.

 L17; Non-standard abbreviations of three minor phytocannabinoids (THCV, CBC, and CBN) should be avoided in the abstract. The full names followed by their abbreviations are strongly recommended.

L37; [1],[2],[3],[4].....change to [1-4].

Line 83; [15],[16].....change to [15, 16]. The authors should revise using such brackets with multiple inserted references.

L100; such as AEA and 2AG.. and Line 114; and CBD....Full names of these cannabinoids should be mentioned.

Line 112; Authors are recommended to define the chemical class of phytocannabinoids and the reason for choosing this class the study.

L123; For this subtitle and all other subtitles, authors should use “capitalized words” as per the journal style.

Line 145-150; Add a suitable reference here.

Figure 1; Screenshots of images with typing cursor should be removed in all figures. The resolution of Figures 1-11 should be increased. Also, the labels’ font size must be increased for more clarity.

L181-187; Add a suitable reference.

L195-199; Add a suitable reference.

L200-203; Add a suitable reference.

L323-329; Add a suitable reference.

Page 19; The “Discussion section”, I suggest that the discussion section be split into several subsections with specific subtitles in a similar way to the results section for more clarity to the reader.

L448; in CB2R, TRPV2, or TRPV4 mRNA expression.... The HUGO Gene Nomenclature Committee (HGNC) endorses the use of italics to denote genes, alleles and RNAs to distinguish them from proteins. So, the authors should check the font style of all genes mentioned in the manuscript and ensure that they follow the HGNC format.

L495-497; Add a suitable reference.

Page 27; “References”; abbreviated journal names should be used, the vol. no. should be in italic font, and the issue numbers should be deleted as per the journal's style. Please follow this style in all references.

For other comments, please find the attached pdf.

Author Response

Abstract; The authors should follow the abstract's style of the journal, which is not a structured one. The authors are recommended to add a conclusion statement at the end of the abstract to highlight the significance of the obtained results. –  The abstract’s style has been changed as per journal format. A conclusion was added to the abstract.

 L17; Non-standard abbreviations of three minor phytocannabinoids (THCV, CBC, and CBN) should be avoided in the abstract. The full names followed by their abbreviations are strongly recommended. – It has been corrected as per your comment. The full names of the phytocannabinoids followed by their abbreviations were incorporated in the abstract.

L37; [1],[2],[3],[4].....change to [1-4].- It was corrected as per your comment.

Line 83; [15],[16].....change to [15, 16]. The authors should revise using such brackets with multiple inserted references. - It was corrected as per your comment.

L100; such as AEA and 2 AG.. and Line 114; and CBD....Full names of these cannabinoids should be mentioned. - It was corrected as per your comment. The full names of AEA, 2AG and CBD have been incorporated in the text. Anandamide (AEA), 2-arachidonoylglycerol (2AG) and cannabidiol (CBD)

Line 112; Authors are recommended to define the chemical class of phytocannabinoids and the reason for choosing this class the study. – It was included in the text.  

Phytocannabinoids represent a consistent group of monoterpenoids characteristic of Cannabis sativa

L123; For this subtitle and all other subtitles, authors should use “capitalized words” as per the journal style. - It was corrected as per your comment.

Line 145-150; Add a suitable reference here. – A suitable reference was added.

Figure 1; Screenshots of images with typing cursor should be removed in all figures. The resolution of Figures 1-11 should be increased. Also, the labels’ font size must be increased for more clarity. - It was corrected as per your comment.

L181-187; Add a suitable reference. - A suitable reference was added.

L195-199; Add a suitable reference. - A suitable reference was added.

L200-203; Add a suitable reference. - A suitable reference was added.

L323-329; Add a suitable reference. - A suitable reference was added.

Page 19; The “Discussion section”, I suggest that the discussion section be split into several subsections with specific subtitles in a similar way to the results section for more clarity to the reader.

The discussion was split into three subsections, including:

3.1. THE EFFECT OF THCV ON LA/LPS INDUCED MACROPHAGES

3.2. THE EFFECT OF CBC ON LA/LPS INDUCED MACROPHAGES

3.3. THE EFFECT OF CBN ON LA/LPS INDUCED MACROPHAGES

L448; in CB2R, TRPV2, or TRPV4 mRNA expression.... The HUGO Gene Nomenclature Committee (HGNC) endorses the use of italics to denote genes, alleles and RNAs to distinguish them from proteins. So, the authors should check the font style of all genes mentioned in the manuscript and ensure that they follow the HGNC format. - It was corrected as per your comments. Th names of genes and mRNA were written in italics.

L495-497; Add a suitable reference. - A suitable reference was added.

Page 27; “References”; abbreviated journal names should be used, the vol. no. should be in italic font, and the issue numbers should be deleted as per the journal's style. Please follow this style in all references.

The "References" section was generated using the "EndNote" software. To determine the referencing style for our citations, we opted for the "Molecules" style from the list of journal styles provided within EndNote.

Reviewer 3 Report

Thank for presentation a study of high interest. Please condiser the following to improve the mansucript.

1) Lines 111-117: Sentences need supporting references.

2) Line 134: better to say "high cytotoxicity" rather "negative impact"

3) Line 166. What does four asterisks signify as seen in the figures? Please define

4) Lines 397-398: According to Figure 10a and 10b the p-values being referred to are the same (p<0.0001) with respect to LPS+ATP. I do not see a difference here. The difference between both concentrations can only be shown if they are compared statistically. If this is the case, I will recommend that you do statistics comparing the treatments among themselves and not just with LPS+ATP only. This will might give some hidden effects which may better explain the results as well as rule out confounding effects on cytokine levels. e.g of the vehicle (methanol) in some of the data 

5) Line 410: These results should come first as it was presented sequentially in the methodology section.

6) Lines 469-472: While these discussions may be inferred, could you references similar studies that compare/contrasts your findings?

7) A significant flaw to the experimental design is lack of two other controls: 

(a) Cell only (non-treated with anything, just cells). The inclusion of cell only will be useful to compare with cell+Methanol (vehicle) to exclude vehicular effects in the downstream effects. It will be great to see the data; and 

 (b) cell + Positive (an anti-inflammatory drug). It is good science to always include a positive control compound as much as possible. Author should include these data.

8) Line 704: (a)Please specify the concentration of seeded macrophages. How many cells per wells were seeded? Which cell culture plates were used for study. (b) What is the rationale for the selection of both concentrations used in the study?

9) Line 712: Give the method for extraction. If a commercial kit was used, please provide manufacturer details.

10) Line 722: Provide the concentration of MTT the cells were exposed to

11) Line 726: Please provide the name of solubilization solution.

12) Line 764: Please give the name of device and details

Author Response

Lines 111-117: Sentences need supporting references. - It was corrected as per your comment.

Supporting references have been included in the text.

2) Line 134: better to say "high cytotoxicity" rather "negative impact" - It was corrected as per your comment. “Negative impact” was replaced with “high cytotoxicity.”

3) Line 166. What does four asterisks signify as seen in the figures? Please define - It was corrected as per your comments.  The definition of four asterisks was included in the caption.

4) Lines 397-398: According to Figure 10a and 10b the p-values being referred to are the same (p<0.0001) with respect to LPS+ATP. I do not see a difference here. The difference between both concentrations can only be shown if they are compared statistically. If this is the case, I will recommend that you do statistics comparing the treatments among themselves and not just with LPS+ATP only. This will might give some hidden effects which may better explain the results as well as rule out confounding effects on cytokine levels. e.g. of the vehicle (methanol) in some of the data - It was corrected as per your comments. The comparison between two doses was added to the figures. That's a fantastic idea to include all the comparisons. In fact, our original intention was to incorporate all the comparisons into the graphs. However, due to the complexity and the need for ample space within the figures to accommodate all these comparisons, we were compelled to include only the comparison with the LA/LPS group. If our aim is to encompass

all the comparisons, the resulting figure would resemble the one depicted below: 

5) Line 410: These results should come first as it was presented sequentially in the methodology section.

A: The location of the MTT results was adjusted to align with its placement in the methodology

6) Lines 469-472: While these discussions may be inferred, could you reference similar studies that compare/contrasts your findings?

The comparisons were integrated into the discussion for CBC and CBN, aligning with earlier findings. It should be mentioned that limited research has explored the anti-inflammatory effects of these three minor phytocannabinoids.

Furthermore, this research examined the response of specific mRNAs/proteins, including STAT3, TYK2, STAT1, Pannexin, P2X7, ADAR, Nlrp3, and cPLA2, to THCV, CBC, and CBN for the first time. As a result, there are no comparable references available to compare the outcomes.

7) A significant flaw to the experimental design is lack of two other controls: 

(a) Cell only (non-treated with anything, just cells). The inclusion of cell only will be useful to compare with cell+Methanol (vehicle) to exclude vehicular effects in the downstream effects. It will be great to see the data; and 

 (b) cell + Positive (an anti-inflammatory drug). It is good science to always include a positive control compound as much as possible. Author should include these data.

A: As highlighted in your comment, incorporating these two control groups could have yielded significant insights. Nevertheless, our experiment's design was centered on investigating the effects of minor phytocannabinoids on induced inflammatory responses in macrophages. Our objective did not encompass examining vehicle effects or drawing a comparison between the phytocannabinoid responses and those induced by an anti-inflammatory drug. Furthermore, the study aimed to unravel the mechanisms underlying the anti-inflammatory effects of these three phytocannabinoids. Given that these phytocannabinoids operate by interacting with the endocannabinoid system (ECS), selecting an appropriate anti-inflammatory drug with a mechanism akin to these phytocannabinoids' actions would have proven unfeasible.

8) Line 704: (a)Please specify the concentration of seeded macrophages. How many cells per wells were seeded? Which cell culture plates were used for study. (b) What is the rationale for the selection of both concentrations used in the study? - It was corrected as per your comment.

  1. a) The cells were grown in cell culture dishes at a concentration of 2-4 x 105 viable cells/mL.
  2. b) In this research, our objective was to examine the anti-inflammatory effects of three minor phytocannabinoids, utilizing two different doses—a targeted optimal dose and a higher dose. Through our optimization trials and in consideration of prior research involving major cannabinoids, we established 5 μM as the optimal concentration. For the high dose, we selected 15 μM for THCV and CBN, as this concentration demonstrated the highest effectiveness without causing noteworthy cytotoxicity. In the case of CBC, its cytotoxic effects at 15 μM compelled us to utilize both the optimal dosage and a lower dosage (2.5 μM).

9) Line 712: Give the method for extraction. If a commercial kit was used, please provide manufacturer details.

A: The methods used for RNA and protein extraction have been mentioned in qPCR and Western blotting sections respectively.

Protein Extraction: the cells were washed twice with cold PBS. Subsequently, the cells were lysed in RIPA buffer and centrifuged at 13,000 rpm for 15 min. The resulting supernatant was transferred to a new microtube, and the protein content was quantified using the Bradford assay. The lysates were then used for western blotting.

RNA extraction: The cellular RNA extraction using TRIzolTM reagent (15596026) as per the manufacturer's instructions.

10) Line 722: Provide the concentration of MTT the cells were exposed to - It was corrected as per your comment.

10 μl (equivalent to 10% of the well's content) of the labeling reagent 3-(4,5-Dimethylthiazol-2-yl)-2,5-diphenyltetrazolium bromide (MTT) into each well.

11) Line 726: Please provide the name of solubilization solution. - It was corrected as per your comment.

Solubilization solution has been prepared by us and is composed of SDS 10% in diluted hydrochloric acid.

12) Line 764: Please give the name of device and details. - It was corrected as per your comment.

The name of device was “SpectraMax i3x Multi-Mode Microplate Reader, Molecular Devices, San Jose, USA”